# Alcohol Consumption Among Spanish Female Adolescents: Related Factors and National Trends 2006–2014

**DOI:** 10.3390/ijerph16214294

**Published:** 2019-11-05

**Authors:** Nazaret Alonso-Fernández, Isabel Jiménez-Trujillo, Valentín Hernández-Barrera, Domingo Palacios-Ceña, Pilar Carrasco-Garrido

**Affiliations:** 1Preventive Medicine and Public Health Teaching and Research Unit, Health Sciences Faculty Universidad Rey Juan Carlos, Alcorcón, 28922 Madrid, Spain; nazaret.alonso@urjc.es (N.A.-F.); isabel.jimenez@urjc.es (I.J.-T.); valentin.hernandez@urjc.es (V.H.-B.); 2Department of Physical Therapy, Occupational Therapy, Rehabilitation and Physical Medicine, Research Group of Humanities and Qualitative Research in Health Science of Universidad Rey Juan Carlos (Hum&QRinHS), Universidad Rey Juan Carlos, Alcorcón, 28922 Madrid, Spain; domingo.palacios@urjc.es

**Keywords:** alcohol drinking, underage drinking, adolescent health, women, epidemiologic studies

## Abstract

Background: The purpose of this study was: (a) to estimate trends over time in the prevalence of alcohol consumption among female adolescents between 2006 and 2014; (b) to identify the factors associated with the probability of consuming alcohol during this period for Spanish female adolescents (14–18 years old). Methods: Spanish nationwide, epidemiological, cross-sectional study on alcohol consumption by adolescent women. We used individualized secondary data retrieved from the 2006 and 2014 Spanish state survey on drug use in secondary education, for a total of 48,676 survey respondents aged 14 to 18 years. Alcohol use was the dependent variable. We also analyzed sociodemographic and educational features, lifestyle habits, perceived health risk for consumption, and perceived availability of substance using logistic regression models. Results: The prevalence of alcohol consumption among female adolescents was 62.35% during the study period. Alcohol consumption increased with age and was more frequent on weekends than on school days. The variables associated with a greater probability of alcohol consumption were tobacco, marijuana (aOR = 2.37; 95% CI: 2.08–2.72), and alcohol consumption by friends (aOR = 7.24; 95% CI: 6.42–8.16). Conclusions: Alcohol consumption by female adolescents in Spain significantly increased from 2006 to 2014. Marijuana and alcohol consumption by friends were associated factors.

## 1. Introduction

It has been widely described in the literature that the early start of alcohol use is a public health concern. In this sense, adolescents who start consuming alcohol before the age of 15 are five times more likely to suffer from alcohol related disorders, and four times more likely to develop alcohol dependence [1]. Several studies in the United States confirm that those teens who start drinking before the age of 15 are more likely to develop a disorder caused by alcohol consumption later in life [1,2]. Europe has the highest alcohol consumption rate per capita in the world. Lithuania stands out as the European country with the highest rate of alcohol consumption, while Sweden is noteworthy for being among those with the lowest consumption [3]. In 2011, per capita consumption in Spanish adults older than 15 was 9.8 liters of alcohol a year. That falls above the average for countries in the organization for economic co-operation and development (OCDE) (9.4 liters of alcohol), and below Germany (11) or Portugal (10.8), but yet above the rate for the USA (8.6), Japan (7.3), or Italy (6.1), and so takes tenth place among European countries [4].

Alcohol is the psychoactive substance most consumed by adolescents in Spain and is considered the gateway to consumption of other substances. After tobacco, alcohol is the second substance most frequently consumed as a first-time experience [5,6]. Previous studies [7,8,9] reported that tobacco and marijuana consumption were significantly associated with greater alcohol use in girls. Moreover, adolescent perception regarding the dangers of alcohol consumption is related to its consumption [7,10,11]. Regarding family and social context, previous studies showed that alcohol use among adolescents is associated with a liberal attitude towards alcohol consumption for parents [12], parents’ patterns of alcohol consumption [9], spending time with friends who drink alcohol [7,8,13], and the presence of peer pressure [14]. Moreover, in respect to school context, it has been established that alcohol consumption is associated with lack of interest in their studies [15], school absenteeism, and poor motivation [16].

The concept of gender refers to the stereotypes, social roles, condition and position acquired, appropriate behaviors, activities, and attributes that each particular society builds and assigns to men and women. Frequently, the introduction of gender sensitivity in studies on social determinant factors in regard to health is primarily focused on populations comprised of women; however, sex and gender are also determinant factors for health among men. Therefore, research on sex-based differences and gender-based disparities in regard to health should take both men and women into consideration and, where possible, separate analysis by sex should be performed. This could aid the understanding of the nature of the multidimensional concept of gender: its social, psychological, and cultural aspects, etc. The influence of gender on drug consumption habits is conditioned by a generational factor. Regarding the adult population, who were mainly brought up according to traditional gender role models, consumption among women is much lower than among men. On the other hand, in the case of adolescents who are receiving a more egalitarian role model education, a trend towards parity of drug consumption habits is now being seen, and the trend has become a full reality for legal psychoactive substances, such as tobacco and alcohol [17,18]. Drug use is still judged differently for women than for men. Therefore, women addicted to drugs face a greater degree of criticism and social rejection which, leads to less family and social support [19]. Among female adolescents, who normally experience puberty earlier than boys, early physical maturation probably contributes to a higher consumption rate for alcohol and other substances [20]. For example, girls may be more sensitive to pressure from peers to fit in and impress others, while male gender role stereotypes regarding alcohol use may be more of a risk factor for boys. Drinking, smoking, and having friends of the opposite sex who engage in criminal behavior can be particularly influential as far as alcohol consumption for female adolescents [21]. The fact that the drugs women consume more commonly are socially accepted and legal has caused them to become invisible to the world of drug dependencies, particularly when it comes to female adolescents, making them an especially vulnerable group [19].

The objectives of this study were: (a) to analyze trends over time in the prevalence of alcohol consumption among female adolescents between 2006 and 2014; and (b) to identify the sociodemographic and education features, self-rated health status, lifestyle habits, perceived health risk for consumption, and perceived availability of substances associated with alcohol use during this period in female adolescents (14–18 years old) in Spain.

## 2. Materials and Methods

### 2.1. Data Source

Cross-sectional epidemiological study, at a national level, on alcohol consumption among in-school female Spanish adolescents generated using individualized secondary data obtained from the encuesta estatal sobre uso de drogas y enseñanzas secundarias (ESTUDES, national survey on drug use in secondary education schools) for 2006, 2008, 2010, 2012, and 2014, carried out by the Spanish ministry of health, social services and equality, and the government delegation of the national drug plan [22]. For the present study, we selected female Spanish adolescents aged 14–18 who were included in the five surveys used. The sample size was 8031 girls for the ESTUDES 2006, 9050 girls for the ESTUDES 2008, 9460 girls for the ESTUDES 2010, 10,088 girls for the ESTUDES 2012, and 12,047 girls for the ESTUDES 2014.

Every two years the spanish national drug plan performs the ESTUDES survey. This is a survey representing students aged 14 to 18 who are in the 3rd and 4th grades of compulsory secondary education, 1st and 4th grades of high school, and intermediate vocational training cycles in Spain in both private and public schools. The survey is carried out every two years in order to assess drug consumption trends and current situations among students aged 14 to 18, using a standardized confidential and self-administered questionnaire which is filled out in writing (paper and pencil) by all students in selected classrooms during a normal class period (45–60 minutes). The field worker goes to the school and explains the rules. The worker remains in the classroom during the whole process and collects the questionnaires after completion.

We used two-stage cluster sampling with the previous random selection of schools as first-stage units, and the classrooms as second-stage units. Subsequently, all students in the selected classrooms were included in the sample in order to simplify the sample design, as well as survey execution and analysis. Details of ESTUDES survey methodology are reported elsewhere [22].

### 2.2. Variables

The dependent variable for this study was the dichotomous answer “yes”, “no” to the question: On how many of the last 30 days did you drink an alcoholic beverage? The type of drinks they consumed, and whether that consumption took place on school days (Monday to Friday) or weekends were used as variables.

The independent variables were classified into three groups: (1) socio-demographic variables, (2) variables related to the study and free time, and (3) consumption of other substances and the environment of the female adolescents. Variables were handled in this study using the same answers’ options provided by the ESTUDES survey [22].

Socio-demographic variables were analyzed: age, nationality (Spanish or foreign), mother’s and father’s occupation (unemployed, employed or inactive), and parents’ education (no formal education, primary education, secondary education, and post-secondary education). 

Regarding the variables related to study and free time, we analyzed class absences within the previous 30 days (dichotomous variable yes/no), repeating grades (dichotomous variable yes/no), and going out at night with friends in the last 12 months (dichotomous variable yes/no) [22].

Variables related to the consumption of other substances and the influence of the environment were: father’s and mother’s alcohol consumption, friends that drink alcohol, and friends that get drunk. We used the questions about tobacco consumption during the previous 30 days (dichotomous variable yes/no) in order to learn about the use of other legal psychoactive substances [22]. For the purpose of finding out about the co-ingestion of illegal psychoactive substances, we included those questions about the use of marijuana, cocaine, sedatives, and other drugs (GHB, ecstasy, amphetamines, hallucinogens, heroine, and inhaled drugs) during the previous month [22]. We also used variables associated with the risk perception of alcohol consumption and the difficulty in obtaining alcohol [22].

### 2.3. Statistical Analysis

The rate of alcohol consumption during the previous 30 days in girls was calculated for each of the five surveys according to the study variables. A bivariate analysis was performed, crossing the independent variables with the “alcohol consumption during the previous 30 days” variable in order to calculate the proportion of subjects who used alcohol according to the study variables for the years 2006, 2008, 2010, 2012, and 2014. The chi-squared test was used for the comparison of proportions.

We assessed the temporal trend for each category in the independent variables. We used a linear trend chi-squared test, considering as significant all values *p* < 0.05.

Subsequently, a multivariate logistic regression analysis was performed to determine the factors associated with the dependent variable (alcohol consumption during the previous 30 days), adjusting for possible confounding factors. Models were obtained for each survey, as well as a joint model for calculating alcohol consumption in 2006 compared to 2008, 2010, 2012, and 2014, respectively. The estimated measure for association was the adjusted Odds ratio (aOR) with its confidence intervals (CI) set to 95%. The selection process to define which independent variable would be part of the multivariate models began with a careful univariable analysis of each variable. Any variable whose univariable test had a *p*-value <0.25 was a candidate for the multivariable model along with all variables of known clinical importance. Following the fit of the multivariable model included an examination of the Wald statistic for each variable, and only those variables that provided significant results and those found to be relevant in the literature were included. We used Hosmer et al. [23]’s goodness of fit test for logistic regression.

All estimations were performed by incorporating the sample design and the survey weighting factors, using ‘svy’ functions (command for the surveys) of the Stata 15.0 program (STATA Corp, College Station, Texas, USA). Statistical signification was set as α = 0.05. 

### 2.4. Ethical Statements

This article does not contain any studies with human participants or animals performed by any of the authors. This article does not need any certificate from the ethics committee, given the nature of the research. All the surveys analyzed were anonymous and dissociated, and contained no recognizable personal information. All content is in accordance with that stated in paragraph two, Section 5 (Orden SAS/3470/2009, December 16th), and does not fall within the assumptions established in Article 2.e (Law 14/2007, June 3rd) concerning biomedical research.

## 3. Results

The sample size of female teenagers aged 14–18 was 8,031 girls for the ESTUDES 2006, 9050 girls for the ESTUDES 2008, 9460 girls for the ESTUDES 2010, 10,088 girls for the ESTUDES 2012, and 12,047 girls for the ESTUDES 2014.

Our results show that the prevalence of consumption among female adolescents residing in Spain is 62.35% during the study period. Alcohol consumption is more frequent on weekends than on school days. The drinks most consumed were beer (16.22%) during the school week and hard liquor (cocktails, mixed drinks, and spirits) (84.18%) on weekends, for the whole study period (Figure 1).

Table 1 shows the rate of consumption according to our socio-demographic variables. Alcohol consumption increases with age for all study years, reaching a rate of 76.59% at age 18. Regarding nationality, female Spanish adolescents present significantly higher levels of alcohol consumption (63.72%) than immigrant girls (51.09%).

Table 2 shows the rate of alcohol consumption for each of the ESTUDES surveys according to the variables related to study and free time. Female adolescents who miss classes (67.93%), have repeated a grade (69.68%), or go out 3 or more nights per week (76.48%), present the highest alcohol consumption rates.

The alcohol consumption rate according to our environmental, psychoactive substance consumption, and risk perception variables can be found in Table 3. The highest alcohol consumption rates for the whole study period were associated with alcohol use among parents and friends. Similarly, alcohol consumption among adolescents was associated with the use of legal and illegal psychoactive drugs for the whole study period. Alcohol consumption rates ranged from 73.49% to 92.57% among those girls who indicated that they consumed marijuana, cocaine, tranquilizers, sedatives, sleeping pills (TSSp), or other drugs.

Table 4 shows the results of the multivariate analysis to identify the variables independently associated with alcohol consumption. Age as a variable is significantly associated with alcohol consumption, with the age of 17 presenting the highest values (aOR = 1.84 [95% CI: 1.68–2.03]). 

Regarding the variables associated with studies and free time, we found that class absence (OR = 1.17 [95% CI: 1.11–1.24]) and going out 3 or more nights per week (aOR = 2.81 [95% CI: 2.48–3.17]) are related to alcohol consumption among female adolescents.

With respect to teenagers’ environment, habitual consumption of alcohol by the mother (aOR = 1.26 [95% CI: 1.10–1.44]), and by all friends is significantly associated with alcohol consumption (aOR = 7.24 [95% CI: 6.42–8.16]), and the latter variable presents the greatest association value found in this study. 

Regarding the consumption of legal and illegal psychoactive substances, female adolescents who smoke were 3.6 times more likely to consume alcohol. Similarly, marijuana consumption in the previous month presents a statistically significant association in adolescents (aOR = 2.37; 95% CI: 2.08–2.72).

Among adolescents who report that they drink 5 or 6 alcoholic beverages on weekends, the perception of several or many problems due to alcohol consumption acts as a protective factor (aOR = 0.62; 95% CI: 0.58–0.66). We also found an association between alcohol consumption and how easy it is to obtain alcohol (aOR = 1.81; 95% CI: 1.62–2.02).

Analysis of trends for alcohol consumption during 2006–2014 revealed that when potential confounders are controlled, we obtained a value aOR = 2.13 95% CI: 1.95–2.34; which means that there has been a significant increase in alcohol consumption from 2004 (57.96%) to 2014 (63.47%) in Spanish adolescent women.

## 4. Discussion

Our results show that there has been an increase in alcohol consumption among female adolescents from 2006 to 2014, reaching prevalence rates of 62.3%. These rates are superior to those reported by Sánchez-Queija et al. [24] in a study on female Spanish adolescents aged 15 to 18, in which 19.6% of girls declared that they drank alcohol frequently, or the results of the health behavior in school-aged children HBSC survey of 2009–2010 in which the rate for consumption in girls was 26% [25]. However, a recent investigation carried out in Portugal to determine the pattern of alcohol consumption in adolescents showed that 86.5% of Portuguese adolescent girls consumed alcohol [26].

Several studies recently published show a decreasing trend in alcohol consumption by adolescents both in Europe and in the United States [27,28]. The research carried out by Looze et al. [29] with adolescents from 28 European and North American countries was designed to assess the trend of weekly alcohol consumption in adolescents from 2002 to 2010. This study showed a reduction in consumption in most countries, mainly Anglo-Saxon countries, and in countries in Northern and Central Europe. However, Mediterranean countries, such as Italy and Portugal, did not present a decreasing trend. 

We observed that as the age of our female adolescents increases, rates of alcohol consumption also increase. In this sense, in a recent study performed with Chilean adolescents aged 10–18 years, female and older students indicated that they were more likely to have consumed alcohol in the 30 days prior to the study than younger students were [7]. Alcohol consumption in teenagers can condition academic performance through a lack of interest in their studies [15], lack of satisfaction, abandonment of school responsibilities, negative attitudes when attending class, school absenteeism, poor motivation, and low self-esteem [16] all negatively impacting learning. All of these have repercussions on students’ present and future lifestyles. Thus, in our study, alcohol consumption was found to be significantly higher in girls who skip classes and go out at night more frequently. Other studies [8,30] confirm that adolescents who reported having average to low school performance were likely to drink more alcohol than those who performed better. Murphy et al. [12], in a study carried out on secondary education students from Southern Ireland, the objective of which was to assess the relationship between alcohol consumption among adolescents and the consumption pattern of their parents, observed that a liberal attitude towards alcohol consumption and an increase in parental consumption levels were linked to risky consumption behaviors in adolescents. One important fact that came to light in our study is that habitual alcohol consumption on the part of mothers and friends represents a risk factor for greater consumption for girls. In a study with boys aged 15–19 years in Brazil, Jorge et al. [9] concluded that the fact that parents consumed alcoholic beverages was a predictor of an increase in drunkenness among adolescents, with mothers demonstrating the greatest influence on the pattern of consumption. Vanassche et al. [31] found that girls whose mothers drank, frequently consumed more alcohol. Furthermore, Ohannessian [32], in a study on American adolescents, found that the mothers’ drinking problems could predict drug consumption for their daughters. Other research shows that parental disapproval of children’s alcohol consumption [8] and parental monitoring (parents who knew where their children were after school) [29] served as protective factors. Several authors, including Villarreal-González et al. [33] in a study on Mexican adolescents aged 12–17, the objective of which was to analyze the relationships among individual, family, school, and social variables and alcohol consumption among adolescents, state that certain factors such as family environment or alcohol consumption on the part of parents can contribute to adolescents starting to consume alcohol. In this line, different investigations suggest the family unit may be ideal for intervening to reduce alcohol use in adolescents in Spain [34].

In terms of their circle of friends, several studies confirm that spending time with friends, especially if they drink alcohol, is associated with a greater likelihood of drinking [7,8,13]. Simons-Morton et al. [14] found that peer pressure was associated with consumption in Finnish girls, but not in boys. Similarly, teaming up with troublesome friends was more strongly associated with consumption for girls as compared to boys. These findings suggest that girls may be more susceptible to peer pressure than are boys. A study with teenagers aged 14–20 from 68 communities in 5 states in the United States reports that one’s group of friends is one of the most influencing variables when it comes to alcohol consumption, since adolescents tend to adjust their alcohol consumption to the people close to them as we have also observed in our study [35]. Several studies [36,37] point out that girls allude to the feeling of safety derived from knowing that if they drink too much their group of friends will take care of them. This feeling of safety contributes to them letting themselves go and drinking alcohol excessively. This becomes evident in the girls’ statements associating alcohol to “having fun”, regardless of the adverse effects of those heavy drinking episodes [35]. Girls see heavy drinking as a way to overcome traditional gender codes. This may be an indicator of changes in women’s position in current society [38]. The influence of gender on drug consumption habits is conditioned by a generational factor. For the adult population, who were mostly brought up with traditional gender role models, consumption among women is much lower than among men. On the other hand, for current adolescents, who are experiencing more egalitarian gender role models, a trend towards parity of drug consumption habits can be observed with parity already reached for legal psychoactive substances, such as tobacco and alcohol, where the lessening of the consumption gap by female teenagers is evident.

We also observed that tobacco and marijuana consumption were significantly associated with greater alcohol use in girls, in line with results obtained in numerous other studies [7,8,9]. Bowden et al. [8] and Jorge et al. [9] specify that smoking acts as a risk factor for alcohol consumption. López et al. [39] found that all cannabis consumers are characterized by also drinking alcohol, thus revealing the existing relationship between the consumption of institutionalized psychoactive substances and the onset of non-institutionalized ones. In the same vein, a Canadian study by Sampasa-Kanyinga et al. [40] found that regular alcohol consumption among students was associated with a greater probability of consuming cannabis (OR = 14.6; 95% CI: 10.8–19.89).

Many adolescents think it is easy to procure alcohol and tobacco, although sales are restricted to persons older than 18. The risk of consumption rises to twenty-four times for poly-drug use [41]. The greater access to alcoholic beverages turned out to be a risk factor for alcohol consumption according to some studies [42].

Alcohol is a legally produced, distributed, and consumed substance, meaning that its use is a normal part of our culture [43]. One of the factors affecting consumption is the easy access to substances and adolescents’ familiarity with them. According to Teixidó-Compañó et al. [44], excessive alcohol consumption is associated with easy access and high availability of alcohol for adolescents. Alcohol sales to minors in stores, as well as advertising for alcoholic beverages in the mass media, promote abusive consumption among teenagers [45,46]. There are studies that demonstrate the relationship between alcohol advertisements and an increase in or start of consumption among adolescents [47,48]. According to these studies, adolescents who have more exposure to alcohol advertising are more likely to start consuming alcohol earlier and drink greater amounts [46]. Suárez-Acevedo et al. [49], in their study with Colombian boys, observed that alcoholic beverage advertising reaches adolescents aged 12–14 via mass media, particularly television, and billboards, flyers, adverts, and the Internet. This contributes to the creation of a relationship between alcohol consumption and socially positive and desired concepts, as well as to standardization of alcohol consumption in that age group. Other studies discuss the influence of exposure to alcohol consumption material on Facebook and consumption among adolescents (the greater the exposure, the greater the consumption) [50].

According to Esnaola Etxaniz [51], the increase in alcoholic beverage consumption among girls may be due to recent social changes, including financial independence and the fact that adolescent girls currently have more freedom and greater access to places where alcoholic beverages are served, places which used to be restricted to boys. Furthermore, girls start consuming alcohol at an earlier age than boys do, because of their tendency to spend time with people older than they are [37].

Adolescent awareness regarding the dangers of alcohol consumption proves to be significantly associated with lower use in several studies [10]. However, other studies, such as one carried out with Chilean children aged 10–18 years, found a greater likelihood of consuming alcohol when the perception of such consumption by adolescents was of little risk to health [7], as occurs in our study with girls who consume a lot of alcohol on weekends. Salamó Avellaneda et al. [11] in their study with Spanish children aged 12–18, who were students attending compulsory secondary education, concluded that adolescents who consume alcohol perceive alcohol as less dangerous than those who do not consume alcohol (*p* < 0.01). Similarly, they observed that girls tend to perceive consumption as more dangerous for boys (*p* < 0.01). In another study [52], the objective of which was to investigate the relationship between risk perception and legal and illegal drug consumption in a Mexican sample of students aged 15–19, a significantly negative correlation was observed between the hazard level of alcohol and the frequency of consumption among girls. Besides, girls’ perception of alcohol’s negative consequences, together with alcohol use as belonging and coping strategies, predicted alcohol consumption in 36.7% [52].

### Limitations

The strengths of our work include: an extensive sample size obtained using a homogeneous methodology over time (2006–2014) and being able to analyze a significant number of socio-demographic variables, and variables associated with alcohol consumption. In addition, there is little scientific literature about alcohol consumption patterns particularly for female adolescents. However, it is also necessary to bear in mind the limitations that this study presents. First, surveys are generally based on self-reported data. Therefore, the information obtained through self-administered questionnaires in classrooms may be subject to recall bias, or to subjects providing socially conditioned answers (lack of veracity of answers on the survey because of having reservations about openly admitting their alcohol use). Thus, the rates used to elaborate alcohol consumption patterns may be underestimated. Finally, it should be taken into account that since this is a descriptive study, no causality can be established in the associations we found. 

The non-response rate, between 9% and 17.1% (ESTUDES), will affect the estimation of the recorded consumption, since those adolescents who refused to participate could share particular characteristics related to consumption, although the direction of the effect cannot be determined [22].

## 5. Conclusions

There has been an increase in alcohol consumption among female adolescents from 2006 to 2014. Girls drink more alcohol on weekends than on school days, and their consumption increases with age. Furthermore, alcohol consumption in adolescents is associated with the use of other drugs (tobacco and marijuana), as well as with factors related to their environment, especially alcohol consumption by friends.

Over time, interventions designed to train parents in skills for communicating clear rules against the use of substances reduce alcohol consumption. Therefore, the need for sensitization about alcohol consumption risks in adolescents becomes evident, together with the need to facilitate the establishment of sensitizing policies, programs, and campaigns effective against alcohol consumption and the negative consequences of these behaviors [53]. To do so, there is a need for action regarding regulations with respect to children’s exposure to advertising, new prevention strategies in social networks and media, and the promoting of healthy leisure time activities in both school and family settings.

## Figures and Tables

**Figure 1 ijerph-16-04294-f001:**
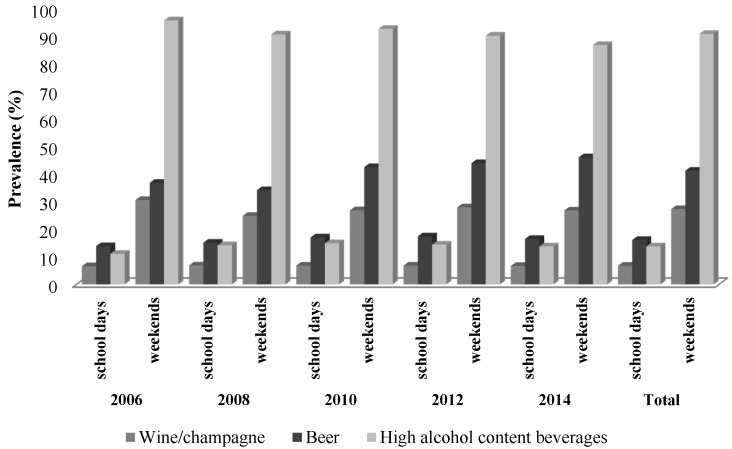
Prevalence of alcohol consumption among Spanish female adolescents aged 14 to 18 years. Consumption patterns and types of beverages ESTUDES surveys 2006, 2008, 2010, 2012, and 2014. ESTUDES: encuesta estatal sobre uso de drogas y enseñanzas secundarias.

**Table 1 ijerph-16-04294-t001:** Prevalence of alcohol consumption among Spanish female adolescents aged 14 to 18 years in Spain, according to sociodemographic variables. ESTUDES surveys 2006–2014.

Variables		2006	2008	2010	2012	2014	Total	*p*-Trend
		N (%)	N (%)	N (%)	N (%)	N (%)	N (%)	
Age ^a.b.c.d.e.f^	14	967 (34.42)	1149(38.72)	1141(35.26)	1022(55.4)	1669(43.06)	5948(40.37)	0.000
	15	1820 (52.83)	2177(56.77)	2247(54.07)	1949(67.4)	2784(57.72)	10,976(57.32)	0.000
	16	2304 (63.96)	2537(64.52)	2970(65.49)	2561(75.84)	3149(69.07)	13,522(67.59)	0.000
	17	2060 (73.29)	2248(70.48)	2356(70.55)	2885(82.11)	3340(76.83)	12,889(74.93)	0.000
	18	880 (74.02)	938(71.77)	745(67.51)	1673(83.7)	1105(80.34)	5342(76.59)	0.000
Nationality^.b.c.d.e.f^	Spanish	7518(58.91)	8280(61.09)	8582(59.08)	9107(75.61)	10,867(64.99)	44,353(63.72)	0.000
	Immigrant	513(46.91)	759(45.96)	868(47.55)	975	1168(52.16)	4283(51.09)	0.000
Occupational status of mother ^e^	Unemployed	224(56.36)	387(55.91)	596(55.57)	1000(74.97)	989(61.84)	3196(62.72)	0.000
	Employed	7511(58.03)	8300(59.9)	8500(57.94)	8728(74.23)	10,510(63.7)	43,549(62.46)	0.000
	Inactive	104(55.95)	92(52.39)	133(62.08)	128(64.24)	159(68.78)	615(61.23)	0.007
Occupational status of father ^e^	Unemployed	123(47.44)	401(57.51)	709(55.26)	1319(72.47)	1102(59.9)	3654(61.94)	0.000
	Employed	7013(58.13)	7689(60.09)	7696(58.21)	7637(74.63)	9588(64.21)	39,622(62.65)	0.000
	Inactive	301(57.31)	315(61.01)	360(56.18)	456(76.99)	465(65.66)	1897(63.6)	0.000
Educational level of parents	No formal	309(59.36)	280(56.9)	196(49.66)	252(73.3)	232(58.44)	1269(59.07)	0.257
	Primary school	2766(59.06)	3164(61.12)	3076(60.8)	3304(76.33)	2728(67.6)	15,038(64.59)	0.000
	Secondary school	1241(61.95)	1246(62.25)	1747(61.79)	1905(76.11)	2806(67.21)	8945(66.21)	0.000
	Higher education	1795(59.71)	2031(62.59)	2326(57.76)	2574(73.52)	3585(62.88)	12312(63.2)	0.000
Total		8031(57.96)	9050(59.41)	9460(57.78)	10,088(74.05)	12,047(63.47)	48,677(62.35)	0.000

a = Significant trend in prevalence of alcohol consumption for the survey ESTUDES2006; b = Significant trend in prevalence of alcohol consumption for the survey ESTUDES2008; c = Significant trend in prevalence of alcohol consumption for the survey ESTUDES2010; d = Significant trend in prevalence of alcohol consumption for the survey ESTUDES2012; e = Significant trend in prevalence of alcohol consumption for the survey ESTUDES2014; f = Significant trend in prevalence of alcohol consumption ESTUDES total.

**Table 2 ijerph-16-04294-t002:** Prevalence of alcohol consumption among Spanish female adolescents aged 14 to 18 years in Spain, according to variables related to study and free time. ESTUDES surveys 2006–2014.

Prevalence in Girls		2006	2008	2010	2012	2014	Total	*p*-Trend
		N (%)	N (%)	N (%)	N (%)	N (%)	N (%)	
Miss classes in the past 30 days ^a^^,^^b,c,d,e,f^	No	2732(47.27)	4268(55.45)	4930(54.1)	4080(67.72)	4879(57.11)	20,889(56.22)	0.000
	Yes	5299(65.61)	4781(63.46)	4530(62.41)	6008(79.07)	7168(68.67)	27,787(67.93)	0.000
Have repeated an academic course ^a.b.c.d.e.f^	No	5276(53.76)	5757(56.56)	6277(54.1)	6990(71.7)	8972(61.4)	33,273(59.46)	0.000
	Yes	2755(68.17)	3293(65.16)	3183(66.74)	3098(79.96)	3075(70.38)	15,404(69.68)	0.000
Go out at night in the last 12 months ^a.b.c.d.e.f^	None	924(26.71)	1000(27.63)	1137(25.85)	1748(47.56)	2248(38.85)	7058(33.7)	0.000
	1–3 nights/month	1991(59.3)	2223(63.61)	2811(63.59)	3443(82.27)	4055(71.71)	14,522(68.79)	0.000
	1–2 nights/week	4154(72.42)	4561(71.45)	4563(72.81)	4065(84.25)	4648(75.91)	21,992(74.97)	0.000
	≥ 3 nights/week	962(73.89)	1265(73.01)	948(73.85)	832(88.67)	1096(77.4)	5104(76.48)	0.000

a = Significant trend in prevalence of alcohol consumption for the survey ESTUDES2006; b = Significant trend in prevalence of alcohol consumption for the survey ESTUDES2008; c = Significant trend in prevalence of alcohol consumption for the survey ESTUDES2010; d = Significant trend in prevalence of alcohol consumption for the survey ESTUDES2012; e = Significant trend in prevalence of alcohol consumption for the survey ESTUDES2014; f = Significant trend in prevalence of alcohol consumption ESTUDES total.

**Table 3 ijerph-16-04294-t003:** Prevalence of alcohol consumption among Spanish female adolescents aged 14 to 18 years, according to variables related to the influence of the environment and use of licit and illicit drugs. ESTUDES surveys 2006–2014.

Variables		2006	2008	2010	2012	2014	Total	*p*-Value
		N (%)	N (%)	N (%)	N (%)	N (%)	N (%)	
Alcohol consumption by father ^a.b.c.d.e.f^	Never	2659(55.35)	3238(55.27)	3562(53.57)	3492(69.58)	3950(58.29)	16,900(58.07)	0.000
	Occasionally	3989(58.33)	4431(61.79)	4729(60.86)	5229(76.8)	6317(66.11)	24,696(64.74)	0.000
	Habitually	1383(62.49)	1381(62.68)	1170(59.87)	1367(76.09)	1779(67.19)	7080(65.47)	0.000
Alcohol consumption by mother ^a.b.c.d.e.f^	Never	3984(54.43)	4752(55.96)	5109(54.27)	4846(71.36)	5323(60.16)	24,014(58.76)	0.000
	Occasionally	3494(61.42)	3705(63.49)	3855(62.76)	4630(76.72)	5988(66.31)	21,670(66.21)	0.000
	Habitually	554(65.26)	593(65.48)	497(60.9)	612(76.8)	737(66.76)	2993(66.94)	0.049
Friends who drink alcohol ^a.b.c.d.e.f^	None of them	275(12.98)	238(11.85)	260(12.19)	460(29.35)	884(25.47)	2117(18.73)	0.000
	Some of them	1725(42.41)	1825	1683(36.69)	2528(63.34)	3202(53.81)	10963	0.000
	All of them	6031(78.66)	6987(77.8)	7517(77.89)	7101(88.03)	7961(83.28)	35,596(81.04)	0.000
Friends who get drunk ^a.b.c.d.e.f^	None of them	1342(30.32)	1281(29.45)	1324(28.39)	1733(49.8)	2517(40.4)	8196(35.41)	0.000
	Some of them	4354(66.11)	5042(68.13)	4971(64.3)	4942(78.62)	5931(70.46)	25,239(69.3)	0.000
	All of them	2335(82.12)	2728(78.27)	3166(79.56)	3414(88.47)	3600(83.03)	15,241(82.38)	0.000
Any cigarette smoking in the past 30 days ^a.b.c.d.e.f^	No	4399(45.75)	4749(47.06)	5229(45.02)	6250(64.85)	8005(55.08)	28631(51.6)	0.000
	Yes	3632(85.66)	4301(83.66)	4231(88.94)	3839(96.3)	4042(90.86)	20,046(88.8)	0.000
Any marijuana use in the past 30 days ^a.b.c.d.e.f^	No	5717(50.35)	6680(52.99)	7164(51.58)	8357(70.59)	10,042(59.78)	37,960(57.09)	0.000
	Yes	2314(92.55)	2370(90.21)	2296(92.5)	1731(96.97)	2005(91.9)	10,717(92.57)	0.034
Any cocaine use in the 30 days ^a.b.c.d.e.f^	No	7854(57.48)	8909(59.09)	9353(57.55)	10,023(73.93)	11,958(63.32)	48,097(62.11)	0.000
	Yes	177(92.49)	141(90.84)	107(90.34)	66(98.88)	89	579(92.28)	0.670
Any tranquilizer. sedative and sleep pill use in the 30 days ^a.b.c.d.e.f^	No	7631(57.57)	8426(58.91)	8791(57.26)	9114(73.06)	10,880(62.32)	44,842(61.56)	0.000
	Yes	400(66.68)	624(67.21)	669(65.75)	974(84.77)	1167(76.66)	3834(73.49)	0.000
Any illicit psychoactive drug use other than marijuana in the last 30 days ^a.b.c.d.e.f^	No	7757(57.3)	8791(58.84)	9237(57.31)	9889(73.74)	11,888(63.25)	47,562(61.93)	0.000
	Yes	275(86.37)	259(88.31)	223(87.49)	199(93.23)	159(86.04)	1115(88.16)	0.511
Perception incidence consumption 5 or 6 cups/week ^a.b.c.d.e.f^	No/few problems	4230(70.09)	5152(72.33)	5080(69.74)	5061(83.11)	5579(76.51)	25101(74.21)	0.000
	Quite a few/many problems	3348(48.06)	3289(47.65)	3449(48.38)	4170(67.68)	5610(56.4)	19,866(53.54)	0.000
	Unknown	453(53.07)	609(50.42)	931(47.56)	858(62.46)	857(49.24)	3709(51.99)	0.486
Perception incidence consumption 1 or 2 cups/day ^a.b.c.d.e.f^	No/few problems	2983(60.75)	3344(64.63)	3402(63.06)	3492(78.89)	4586(70.12)	17,807(67.33)	0.000
	Quite a few/many problems	4520(56.75)	5015(57.28)	5117(56.62)	5782(73.25)	6548(61.46)	26,982(60.9)	0.000
	Unknown	529(53.88)	691(53.04)	941(48.52)	814(62.44)	913(51.06)	3888(53.15)	0.647
Perception incidence consumption 5 or 6 cups/day ^a.b.c.d.e.f^	No/few problems	615(66.81)	777(69.57)	833(70.16)	817(82.68)	767(71.19)	3809	0.000
	Quite a few/many problems	6813(57.22)	7498(58.92)	7555(57.78)	8328(74.66)	10247(64.19)	40,440(62.38)	0.000
	Unknown	603(58.64)	775(55.77)	1072(50.83)	944(63.69)	1033(53.26)	4427(55.7)	0.618
Perceived availability of alcohol ^a.b.c.d.e.f^	Impossible/very difficult to obtain	326(27.5)	321(25.71)	258(26.93)	378(46.16)	495(35.18)	1776(31.65)	0.000
	Easy/very easy to obtain	7375(62.15)	8351(64.3)	8658(62.46)	9098(78.89)	10,721(69.87)	44,202(67.39)	0.000

a = Significant trend in prevalence of alcohol consumption for the survey ESTUDES2006; b = Significant trend in prevalence of alcohol consumption for the survey ESTUDES2008; c = Significant trend in prevalence of alcohol consumption for the survey ESTUDES2010; d = Significant trend in prevalence of alcohol consumption for the survey ESTUDES2012; e = Significant trend in prevalence of alcohol consumption for the survey ESTUDES2014; f = Significant trend in prevalence of alcohol consumption ESTUDES total.

**Table 4 ijerph-16-04294-t004:** Factors associated with alcohol consumption in Spanish female adolescents aged 14 to 18 years. ESTUDES surveys 2006–2014.

Variables		2006	2008	2010	2012	2014	Total
		aOR (95%CI)	aOR (95%CI)	aOR (95%CI)	aOR (95%CI)	aOR (95%CI)	aOR (95%CI)
		1.20(1.12–1.29)	1.17(1.10–1.26)	1.17(1.09–1.23)	1.16(1.09–1.24)	1.23(1.17–1.24)	1.19(1.15–1.22)
**Miss classes in the past 30 days**	No	1	N.S.	N.S.	1	1	1
	Yes	1.22(1.06–1.42)	N.S.	N.S.	1.41(1.22–1.62)	1.26(1.13–1.40)	1.17(1.11–1.24)
**Go out at night in the last 12 months**	None	1	1	1	1	1	1
	1–3 nights/month	2.37(1.94–2.90)	2.54(2.09–3.08)	2.87(2.46–3.36)	2.91(2.44–3.47)	2.19(1.91–2.50)	2.49(2.31–2.68)
	1–2 nights/week	3.37(2.78–4.06)	2.75(2.29–3.30)	3.46(2.97–4.02)	2.68(2.24–3.20)	2.46(2.15–2.82)	2.86(2.66–3.08)
	≥ 3 nights/week	3.45(2.53–4.71)	2.92(2.28–3.75)	2.88(2.22–3.74)	2.95(2.07–4.20)	2.25(1.76–2.87)	2.81(2.48–3.17)
**Alcohol consumption by mother**	Never	1	1	1	N.S.	1	1
	Occasionally	1.26(1.06–1.50)	1.24(1.06–1.45)	1.29(1.12–1.49)	N.S.	1.14(1.01–1.30)	1.21(1.13–1.29)
	Habitually	1.54(1.10–2.17)	1.62(1.20–2.19)	1.33(1.02–1.78)	N.S.	1.03(0.79–1.33)	1.26(1.10–1.44)
**Friends who drink alcohol**	None of them	1	1	1	1	1	1
	Some of them	3.04(2.25–4.11)	3.03(2.22–4.13)	2.84(2.16–3.74)	2.68(2.08–3.48)	2.09(1.75–2.50)	2.47(2.22–2.76)
	All of them	8.57(6.21–11.82)	9.03(6.50–12.52)	10.96(8.22–14.63)	6.89(5.13–9.26)	5.09(4.14–6.26)	7.24(6.42–8.16)
**Friends who get drunk**	None of them	N.S.	1	1	N.S.	N.S.	1
	Some of them	N.S.	1.45(1.22–1.73)	1.26(1.08–1.48)	N.S.	N.S.	1.19(1.10–1.28)
	All of them	N.S.	1.27(1.01–1.60)	1.12(0.92–1.38)	N.S.	N.S.	1.11(1.01–1.23)
**Any cigarette smoking in the past 30 days**	No	1	1	1	1	1	1
	Yes	2.81(2.31–3.41)	2.82(2.34–3.39)	4.08(3.44–4.84)	6.72(5.22–8.66)	4.02(3.34–4.84)	3.60(3.30–3.92)
**Any marijuana use in the past 30 days** **s**	No	1	1	1	1	1	1
	Yes	3.06(2.28–4.10)	2.30(1.74–3.04)	2.84(2.18–3.69)	2.48(1.58–3.89)	1.50(1.14–1.98)	2.37(2.08–2.72)
**Perception incidence consumption 5 or 6 cups/week**	No/few problems	1	1	1	1	1	1
	Quite a few/many problems	0.64(0.55–0.75)	0.53(0.46–0.61)	0.66(0.58–0.75)	0.60(0.52–0.70)	0.64(0.58–0.72)	0.62(0.58–0.66)
	Unknown	0.71(0.43–1.18)	0.75(0.53–1.05)	0.96(0.68–1.36)	0.78(0.53–1.14)	0.66(0.49–0.90)	0.79(0.68–0.93)
**Perceived availability of alcohol**	Impossible/very difficult to obtain	1	1	1	1	1	
	Easy/very easy to obtain	1.60(1.19–2.15)	1.87(1.47–2.39)	1.55(1.21–2.00)	1.82(1.37–2.42)	1.72(1.40–2.12)	1,81(1,62–2,02)

aOR = adjusted OR; CI = confidence interval; N.S. = non-significant trend in prevalence of alcohol consumption.

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
