# Peer review of "Alcohol Consumption Among Spanish Female Adolescents: Related Factors and National Trends 2006–2014"

_ijerph, 2019, doi:10.3390/ijerph16214294_

Round 1
Reviewer 1 Report
The research presented in the manuscript might be interesting for the readers of the journal, but some improvements in the manuscript must be done:
1) the introduction part of the manuscript lacks analysis of the important independent variables presented in the research part of the manuscript;
2) more information concerning process of making dichotomous variables in the section 2.2. of the manuscript must be provided; particular questions of measuring consumption of other substances and the influence of the environment must be provided, as now it is not possible to understand how these variables were done;
3) more information about regression models must be provided (Adj. R squares of the overall models) and interpreted in the discussion part of the manuscript, as this information might be useful to understand about explanation of the variance of dependent variable;
4) Age is not an ordinal variable, so changes in regression models should be done;
5) notes in the end of all tables are not clear - what does it mean 'significant association' (it is not clear what is related with what);
6) analysis and interpretation of the time trends must be presented in the discussion part, because it is implied in the title of the manuscript.
Author Response
RESPONSE LETTER
International Journal of Environmental Research and Public Health
Manuscript ID: ijerph-627240
Alcohol consumption among Spanish female adolescents: Related factors and national trends 2006-2014.
Reviewers' comments:
REVIEWER 1:
The research presented in the manuscript might be interesting for the readers of the journal, but some improvements in the manuscript must be done:
1) the introduction part of the manuscript lacks analysis of the important independent variables presented in the research part of the manuscript;
Response: We agree with reviewer. We included new information at introduction section:
Previous studies [7-9] reported that tobacco and marijuana consumption were significantly associated with greater alcohol use in girls. Moreover, adolescent perception regarding the dangers of alcohol consumption is related to its consumption [7,10,11]. Regarding family and social context, previous studies showed that alcohol use among adolescents is associated with a liberal attitude towards alcohol consumption for parents [12], parents´ patterns of alcohol consumption [9], spending time with friends who drink alcohol [7,8,13], and the presence of peer pressure [14]. Moreover, in respect of school context, it have been established that alcohol consumption is associated with lack of interest in their studies [15], school absenteeism, and poor motivation [16].
Also, we need to change references order.
2) more information concerning process of making dichotomous variables in the section 2.2. of the manuscript must be provided; particular questions of measuring consumption of other substances and the influence of the environment must be provided, as now it is not possible to understand how these variables were done;
Response: In the present study, we used the questions and their respective answer's options as they were described in the respective surveys (Plan Nacional sobre Drogas, 2011). No stratification was performed in order dichotomize any variables. The authors did not participate in the development of the surveys. ESTUDES is a Spanish Public National Survey performed periodically so, unfortunately, we are not able to provide any information about particular questions of the measuring consumption of other substances and the influence of the environment.
In order to clarify the independent variables, we added the following : Variables were handled in this study using the same answers’ options provided by the ESTUDES Survey [22].
We included at methods section:
Details of ESTUDES Survey methodology was reported elsewhere [22].
Variables were handled in this study using the same answers’ options provided by the ESTUDES Survey [22].
Regarding the variables related to the study and free time we analyzed class absences within the previous 30 days (dichotomous variable yes/no), repeating grades (dichotomous variable yes/no), and going out at night with friends in the last 12 months (dichotomous variable yes/no) [22].
Variables related to the consumption of other substances and the influence of the environment were: father’s and mother’s alcohol consumption, friends that drink alcohol, and friends that get drunk. We used the questions about tobacco consumption during the previous 30 days (dichotomous variable yes/no) in order to learn about the use of other legal psychoactive substances [22]. For the purpose of finding out about the co-ingestion of illegal psychoactive substances, we included those questions about the use of marijuana, cocaine, sedatives and other drugs (GHB, ecstasy, amphetamines, hallucinogens, heroine and inhaled drugs) during the previous month [22]. We also used variables associated with the risk perception of alcohol consumption and the difficulty in obtaining alcohol [22].
Reference: 22. Plan Nacional sobre Drogas. Encuesta estatal sobre uso de drogas en estudiantes de enseñanzas secundarias. ESTUDES, 2010 [State survey on drug use in secondary school students. ESTUDES, 2010]. Delegación del gobierno para el plan nacional sobre drogas, Madrid, 2011.
3) more information about regression models must be provided (Adj. R squares of the overall models) and interpreted in the discussion part of the manuscript, as this information might be useful to understand about explanation of the variance of dependent variable;
Response: The selection process begin with a careful univariable analisys of each variable, any variable whose univariable test has a p-value <0.25 is a candidate for the multivariable model along with all variables of know clinical importance, following the fit of the multivariable model, include an examination of the Wald statistic for each variable and only those variables that provided significant results and those found to be relevant in the literature were included. We do not use the R2 statistic for logistic regression, because it is almost always rather low, it is rarely used. See Hosmer et al (2013, page 164) for details (they themselves recommend not using this method), for assessing the fit of the model, we used Hosmer et al (2013) goodness of fit test, and all model were not significant.
Hosmer Jr, D. W.; Lemeshow, S.; Sturdivant, R. X. Applied Logistic Regression. Third Edition. Wiley, 2013: 153-225. doi: 10.1002/9781118548387
This information was included at methods section:
The selection process to define which independent variable would be part of the multivariate models began with a careful univariable analisys of each variable. Any variable whose univariable test had a p-value <0.25 was a candidate for the multivariable model along with all variables of known clinical importance. Following the fit of the multivariable modelincluded an examination of the Wald statistic for each variable and only those variables that provided significant results and those found to be relevant in the literature were included. We used Hosmer et al [23] goodness of fit test for logistic regression.
Reference number 23: Hosmer Jr, D. W.; Lemeshow, S.; Sturdivant, R. X. Applied Logistic Regression. Third Edition. Wiley, 2013: 153-225. doi: 10.1002/9781118548387
4) Age is not an ordinal variable, so changes in regression models should be done;
Response: We agree. The changes have been made and have been introduced in the paper at table 4.
|
2006 |
2008 |
2010 |
2012 |
2014 |
Total |
||
|
|
|
aOR (95%CI) |
aOR (95%CI) |
aOR (95%CI) |
aOR (95%CI) |
aOR (95%CI) |
aOR (95%CI) |
|
Age |
1.20(1.12-1.29) |
1.17(1.10-1.26) |
1.17(1.09-1.23) |
1.16(1.09-1.24) |
1.23(1.17-1.24) |
1.19(1.15-1.22) |
5) notes in the end of all tables are not clear - what does it mean 'significant association' (it is not clear what is related with what);
Response: The meaning is “Significant trend in Prevalence of alcohol consumption for the survey ESTUDES”.
We included: “a=Significant trend in Prevalence of alcohol consumption for the survey ESTUDES2006; b=Significant trend in Prevalence of alcohol consumption for the survey ESTUDES2008; c=Significant trend in Prevalence of alcohol consumption for the survey ESTUDES2010; d=Significant trend in Prevalence of alcohol consumption for the survey ESTUDES2012; e=Significant trend in Prevalence of alcohol consumption for the survey ESTUDES2014; f=Significant trend in Prevalence of alcohol consumption ESTUDES Total.” At table 1,2 and 3.
Also, we included: “aOR=adjusted OR; CI=confidence interval; N.S.=non-significant trend in Prevalence of alcohol consumption.” at table 4.
6) analysis and interpretation of the time trends must be presented in the discussion part, because it is implied in the title of the manuscript.
RESPONSE: We agree with the comment. This question has been clarified in the Results section. The following text has been added:
Analysis of trends for alcohol consumption during 2006-2014 revealed that when potential confounders are controlled, we obtained a value aOR=2.13 95% CI: 1.95-2.34; which means that there has been a significant increase in alcohol consumption from 2004 (57.96%) to 2014 (63.47%), in Spanish adolescent women.

Reviewer 2 Report
First place thank the authors the good work performed and the good synthesis of the state of science collected
From line 55 to 69, the authors refer to the genre. It would be interesting to pick up some kind of theory that the woman's papael addressed. Theory of cultural force or aspects collected by Margaret Mead or Simone de Beavuoir.
On line 259 the authors discuss consumption and gender. A broader view of the possibilities would be necessary.Are the most consuming women of alcohol to gain power in society? Why?Why now that consumption rates are decreasing?
Author Response
RESPONSE LETTER
International Journal of Environmental Research and Public Health
Manuscript ID: ijerph-627240
Alcohol consumption among Spanish female adolescents: Related factors and national trends 2006-2014.
REVIEWER 2
First place thank the authors the good work performed and the good synthesis of the state of science collected
From line 55 to 69, the authors refer to the genre. It would be interesting to pick up some kind of theory that the woman's papael addressed. Theory of cultural force or aspects collected by Margaret Mead or Simone de Beavuoir.
RESPONSE: Thank you for this comment. The following text has been added in the introduction section:
The concept of gender refers to the stereotypes, social roles, condition and position acquired, appropriate behaviors, activities and attributes that each particular society builds and assigns to men and women. Frequently, the introduction of gender sensitivity in studies on social determinant factors in regard to health is primarily focused on populations comprised of women; however, sex and gender are also determinant factors for health among men. Therefore, research on sex-based differences and gender-based disparities in regard to health should take both men and women into consideration and, where possible, separate analysis by sex should be performed. This could aid the understanding of the nature of the multidimensional concept of gender: its social, psychological and cultural aspects, etc.
On line 259 the authors discuss consumption and gender. A broader view of the possibilities would be necessary. Are the most consuming women of alcohol to gain power in society? Why?Why now that consumption rates are decreasing?
RESPONSE: Thank you for this comment. The following text has been added in the discussion section:
The influence of gender on drug consumption habits is conditioned by a generational factor. For the adult population, who were mostly brought up with traditional gender role models, consumption among women is much lower than among men.
On the other hand, for current adolescents, who are experiencing more egalitarian gender role models, a trend towards parity of drug consumption habits can be observed with parity already reached for legal psychoactive substances, such as tobacco and alcohol, where the lessening of the consumption gap by female teenagers is evident

Reviewer 3 Report
This is an interesting and important review discussing trends in alcohol consumption among Spanish female ‘adolescents’.
Drawing on secondary data retrieved from Spanish school reviews in 2006 and 2014, the analysis includes consideration of sociodemographic and educational features, lifestyle habits, perceived health risks for consumption and perceived availability. The authors correctly note the limitations due to self-reporting and how and where the data is gathered, ie in schools, where it has been suggested from other studies that there might be under-reporting of unsanctioned, including illicit behaviours.
I have only a few suggestions for amendments.
The paper is well constructed, with evidence presented in a coherent, clear way.
Some editing could usefully revise the English used – eg line 60-61,
Some of the arguments require more explanation and validation – eg line 64 that ‘physical maturation’ leads girls to consume more than boys. I would also have liked some exploration of why the reported use appears so much higher than neighbouring Portugal. Studies from further afield, eg Brazil, are cited for comparison.
From line 143 the argument is made that there is no need for ethical approval. I would have thought that there would be such a requirement, in relation to how the data is going to be analysed, used and disseminated.
It would be useful just to make sure that there is consistent use of terms throughout – eg adolescent.
Maybe these two papers would also be useful:
Goldberg-Looney, L.D. et al. Adolescent drinking in Spain: Family relationship quality, rules, communication, and behaviors, in Children and Youth Services Review
Volume 58, November 2015, Pages 236-243
Goldberg-Looney, L.D. et al. Adolescent Alcohol Use in Spain: Connections with Friends, School, and Other Delinquent Behaviors, in Frontiers in Psychology, 2016; 7: 269. https://www.ncbi.nlm.nih.gov/pmc/articles/PMC4776124/
I hope that’s useful.
Author Response
RESPONSE LETTER
International Journal of Environmental Research and Public Health
Manuscript ID: ijerph-627240
Alcohol consumption among Spanish female adolescents: Related factors and national trends 2006-2014.
REVIEWER 3
This is an interesting and important review discussing trends in alcohol consumption among Spanish female ‘adolescents’.
Drawing on secondary data retrieved from Spanish school reviews in 2006 and 2014, the analysis includes consideration of sociodemographic and educational features, lifestyle habits, perceived health risks for consumption and perceived availability. The authors correctly note the limitations due to self-reporting and how and where the data is gathered, ie in schools, where it has been suggested from other studies that there might be under-reporting of unsanctioned, including illicit behaviours.
I have only a few suggestions for amendments.
The paper is well constructed, with evidence presented in a coherent, clear way.
Some editing could usefully revise the English used – eg line 60-61,
RESPONSE: Thank you for this comment. The text has been reviewed by a native English speaker.
Some of the arguments require more explanation and validation – eg line 64 that ‘physical maturation’ leads girls to consume more than boys.
RESPONSE: We agree with the comment. The paragraph has been modified. The following text has been added:
For example, girls may be more sensitive to pressure from peers to fit in and impress others, while male gender role stereotypes regarding alcohol use may be more of a risk factor for boys.
I would also have liked some exploration of why the reported use appears so much higher than neighbouring Portugal. Studies from further afield, eg Brazil, are cited for comparison.
RESPONSE: Thank you for this comment. The following text has been added in the discussion section and we have added a new reference. All references have been re-numbered.
“However, a recent investigation carried out in Portugal to determine the pattern of alcohol consumption in adolescents showed that 86.5% of Portuguese adolescents girls consumed alcohol”
[26] Gonçalves IA, Carvalho AAS. Pattern of Alcohol Consumption by Young People from North Eastern Portugal. Open Med (Wars) 2017,12, 494-500. doi: 10.1515/med-2017-0068.
From line 143 the argument is made that there is no need for ethical approval. I would have thought that there would be such a requirement, in relation to how the data is going to be analysed, used and disseminated.
RESPONSE: Thank you for this comment In accordance with Spanish legislation, Ethics Committee approval was not necessary, since the databases were obtained from the Spanish Ministry of Health website, where they are publicly available.
It would be useful just to make sure that there is consistent use of terms throughout – eg adolescent.
RESPONSE: Following your suggestion the term adolescents has been applied in the manuscripts
Maybe these two papers would also be useful:
Goldberg-Looney, L.D. et al. Adolescent drinking in Spain: Family relationship quality, rules, communication, and behaviors, in Children and Youth Services Review.Volume 58, November 2015, Pages 236-243
Goldberg-Looney, L.D. et al. Adolescent Alcohol Use in Spain: Connections with Friends, School, and Other Delinquent Behaviors, in Frontiers in Psychology, 2016; 7: 269. https://www.ncbi.nlm.nih.gov/pmc/articles/PMC4776124/
RESPONSE: Following your suggestion these references has been added in the discussion section.
We included:
Goldberg-Looney, L.D.; Sánchez-SanSegundo, M.; Ferrer-Cascales, R.; Smith, E.R.; Albaladejo-Blazquez, N.; Perrina, P.B. Adolescent drinking in Spain: Family relationship quality, rules, communication, and behaviors. Children and Youth Services Review 2015, 58, 236-243. doi: 10.1016/j.childyouth.2015.09.022
Goldberg-Looney, L.D.; Sánchez-SanSegundo, M.; Ferrer-Cascales, R.; Albaladejo-Blazquez, N.; Perrin, P.B. Adolescent Alcohol Use in Spain: Connections with Friends, School, and Other Delinquent Behaviors. Front Psychol 2016, 7, 269. doi: 10.3389/fpsyg.2016.00269.
